# Polygenic risk-tailored screening for prostate cancer: A benefit–harm and cost-effectiveness modelling study

**Tom Callender**[1]*, **Mark Emberton**[2], **Steve Morris**[1], **Ros Eeles**[3], **Zsofia Kote-Jarai**[3], **Paul D. P. Pharoah**[4], **Nora Pashayan**[1]

**1** Department of Applied Health Research, Institute of Epidemiology & Health Care, University College London, London, United Kingdom, **2** Faculty of Medical Sciences, School of Life & Medical Sciences, University College London, London, United Kingdom, **3** The Institute of Cancer Research, London, United Kingdom, **4** Departments of Oncology, and Public Health and Primary Care, Strangeways Research Laboratory, University of Cambridge, Cambridge, United Kingdom

* tom.callender@ucl.ac.uk

**Data Availability Statement:** All relevant data are within the manuscript and its Supporting Information files. The data files used in this

## Abstract

### Background

The United States Preventive Services Task Force supports individualised decision-making for prostate-specific antigen (PSA)-based screening in men aged 55–69. Knowing how the potential benefits and harms of screening vary by an individual's risk of developing prostate cancer could inform decision-making about screening at both an individual and population level. This modelling study examined the benefit–harm tradeoffs and the cost-effectiveness of a risk-tailored screening programme compared to age-based and no screening.

### Methods and findings

A life-table model, projecting age-specific prostate cancer incidence and mortality, was developed of a hypothetical cohort of 4.48 million men in England aged 55 to 69 years with follow-up to age 90. Risk thresholds were based on age and polygenic profile. We compared no screening, age-based screening (quadrennial PSA testing from 55 to 69), and risk-tailored screening (men aged 55 to 69 years with a 10-year absolute risk greater than a threshold receive quadrennial PSA testing from the age they reach the risk threshold). The analysis was undertaken from the health service perspective, including direct costs borne by the health system for risk assessment, screening, diagnosis, and treatment. We used probabilistic sensitivity analyses to account for parameter uncertainty and discounted future costs and benefits at 3.5% per year. Our analysis should be considered cautiously in light of limitations related to our model's cohort-based structure and the uncertainty of input parameters in mathematical models. Compared to no screening over 35 years follow-up, age-based screening prevented the most deaths from prostate cancer (39,272, 95% uncertainty interval [UI]: 16,792–59,685) at the expense of 94,831 (95% UI: 84,827–105,630) overdiagnosed cancers. Age-based screening was the least cost-effective strategy studied. The greatest number of quality-adjusted life-years (QALYs) was generated by risk-based

analysis are also available with the code at https://github.com/callta/precision_screening_pca.

**Funding:** The authors received no specific funding for this work.

**Competing interests:** The authors have declared that no competing interests exist.

**Abbreviations:** AUC, area under the curve; CI, confidence interval; ICER, incremental cost-effectiveness ratio; NHS, National Health Service; NICE, National Institute for Health and Care Excellence; NMB, net monetary benefit; PSA, prostate-specific antigen; QALY, quality-adjusted life-year; SE, standard error; SNP, single-nucleotide polymorphism; UI, uncertainty interval.

screening at a 10-year absolute risk threshold of 4%. At this threshold, risk-based screening led to one-third fewer overdiagnosed cancers (64,384, 95% UI: 57,382–72,050) but averted 6.3% fewer (9,695, 95% UI: 2,853–15,851) deaths from prostate cancer by comparison with age-based screening. Relative to no screening, risk-based screening at a 4% 10-year absolute risk threshold was cost-effective in 48.4% and 57.4% of the simulations at willingness-to-pay thresholds of GBP£20,000 (US$26,000) and £30,000 ($39,386) per QALY, respectively. The cost-effectiveness of risk-tailored screening improved as the threshold rose.

## Conclusions

Based on the results of this modelling study, offering screening to men at higher risk could potentially reduce overdiagnosis and improve the benefit–harm tradeoff and the cost-effectiveness of a prostate cancer screening program. The optimal threshold will depend on societal judgements of the appropriate balance of benefits–harms and cost-effectiveness.

## Author summary

### Why was this study done?

- Prostate cancer screening using prostate-specific antigen has been shown to lead to a reduction in prostate-cancer–specific mortality at the expense of overdiagnosis and overtreatment.

- Genome-wide association studies have identified more than 160 common genetic variants that, when combined together as a polygenic risk score, might be used to develop a tailored screening programme for prostate cancer.

- The proportion of men overdiagnosed has been shown to vary by polygenic risk; therefore, a risk-tailored screening based on age and polygenic risk profile may improve the balance of benefits and harms of a screening programme for prostate cancer.

### What did the researchers do and find?

- We developed a mathematical model that simulated hypothetical cohorts of 4.48 million men aged 55 to 69 in England.

- Using this model, we analysed the balance of benefits and harms in terms of prostate-cancer–specific mortality reduction against overdiagnosis, as well as the cost-effectiveness, of the introduction of a risk-tailored screening programme for prostate cancer based on age and polygenic risk.

- We compared risk-tailored screening to age-based screening and no screening.

### What do these findings mean?

- Based on this model, we show that a polygenic risk-tailored screening programme might reduce overdiagnosis, maintain the mortality benefits of age-based screening, and improve the cost-effectiveness of a screening programme for prostate cancer.

- The ideal threshold for risk-tailored screening will depend on societal judgement of the tradeoff between the benefits and harms of screening.

- Future prospective evaluations are needed to verify these findings.

## Introduction

Screening with a prostate-specific antigen (PSA) test could reduce death from prostate cancer in some men but at the cost of substantial numbers overdiagnosed, as well as false positive results [1]. Overdiagnosed cancers are the screen-detected cancers that would not otherwise have come to light during an individual's lifetime [2], whose diagnosis and related treatment can lead to avoidable physical and psychological harms whilst also incurring an economic cost [3]. The updated 2018 guidelines of the US Preventive Services Taskforce recommend consideration of screening for certain at risk men between the ages of 55 and 69 [1]. Understanding the variation in the potential benefits and harms of screening by an individual's risk of developing prostate cancer could inform decision-making about screening at both the individual and population level.

Genome-wide association studies have identified common susceptibility loci that together explain approximately 37% of the familial relative risk of prostate cancer [4]. Individually, these loci have little clinical significance [5]. However, together they define a risk distribution with a variance of 0.68 and area under the curve (AUC) of 0.72 that can be used to stratify individuals into groups at higher and lower risk of developing prostate cancer [2,6], enabling tailored screening by risk group. Individuals in the first and 99th percentiles of the polygenic risk distribution have relative risks of developing prostate cancer of 0.09 and 5.52, respectively, compared to the population mean. Almost 49% of prostate cancers occur in those men in the highest 20% of the polygenic risk distribution, and only 7% occur amongst those in the lowest 20% of the risk distribution (Fig C in S1 Appendix). Avoiding screening of those at lower risk may consequently reduce the harms of screening without a commensurate loss of benefit. The proportion of prostate cancers overdiagnosed varies inversely with polygenic risk, with almost 50% lower overdiagnosis in men in the highest quartile compared with the lowest quartile of polygenic risk [2]. This approach to screening would require genotyping of all men for the purpose of risk assessment. However, the additional costs of population-wide genotyping may be offset by lower levels of overtreatment.

The aims of this study were to assess the balance of benefit and harms, as well as the cost-effectiveness, of the introduction of a polygenic risk-tailored screening programme for prostate cancer.

## Methods

### Model structure

We developed a life-table model (Fig A in S1 Appendix) to simulate cohorts of men under 3 scenarios: no screening, age-based screening with PSA, and risk-tailored screening (henceforth, precision screening). Our life table is a cohort-based Markov model that estimates the age-specific incidence of prostate cancer, deaths from prostate cancer, and deaths from other causes. Each hypothetical cohort consisted of 4.48 million men aged 55 to 69, representing the

mean population of men of this age in England in 2013–2016 [7]. All cohorts were assumed to be prostate-cancer–free on entering the model and were followed up to the age of 90.

Age-based screening involved PSA testing every 4 years between the ages of 55 and 69, reflecting the screening interval used in the core analyses of the European Randomized Study of Screening for Prostate Cancer [8]. In the precision screening cohort, we estimated the 10-year absolute risk of developing prostate cancer based on age and polygenic profile. From the age of 55 to 69, this cohort of men started quadrennial PSA screening at the age at which they reached a specified risk threshold, which was varied between a 2% and 10% 10-year absolute risk of developing prostate cancer. We set a PSA cutoff of 3.0 ng/mL for suspected prostate cancer requiring further assessment for both the age-based and precision screening cohorts.

### Precision screening

The distribution of polygenic risk on a relative risk scale in the population is log-normal, assuming a log-additive interaction between loci [6]. The variance of the polygenic risk distribution was calculated as 0.68, based on known prostate cancer susceptibility variants (see S1 Appendix for further details) [6,9].

We calculated the log relative risk of developing prostate cancer for each risk threshold respective to the background 10-year absolute risk of developing prostate cancer in the absence of screening. By applying the log relative risk of developing prostate cancer to the polygenic risk distribution, we then determined the proportion above the risk threshold and the proportion of all cases of prostate cancer accounted for amongst those above the threshold. From this, we derived the relative risk of developing prostate cancer in those who were screened or not screened.

### Model parameters

Model parameters are shown in Table 1. We used Office for National Statistics data and Dev-Can software to calculate the age-specific incidence of prostate cancer, mortality from prostate cancer, and mortality from other causes, accounting for competing risks (Table B in S1 Appendix) [10]. The incidence of prostate cancer in the nonscreened cohorts was adjusted by 10% to reflect the estimated proportion of cancers that are diagnosed as a result of opportunistic screening in England (see S1 Appendix for further details). Costs were estimated from the perspective of the National Health Service (NHS) in 2016 prices using Unit Costs of Health and Social Care and NHS Reference Costs, which have been previously validated for use in costing prostate cancer care (Table 1 and Table A in S1 Appendix) [11–13]. We determined the proportion receiving different treatments by stage at diagnosis from the National Cancer Registration and Analysis Service, the National Prostate Cancer Audit, and National Institute for Health and Care Excellence (NICE) guidelines [14–16]. The cost of polygenic risk stratification was based on an empirical estimate.

We developed utility weights for those with prostate cancer based on treatment modality (see S1 Appendix for further details). We estimated the number of overdiagnoses by multiplying the number of screen-detected cases by the age-specific proportion estimated to be at risk of overdiagnosis [17].

### Model outputs

We calculated costs, the number of quality-adjusted life-years (QALYs), life-years, prostate cancer cases, overdiagnoses, deaths from prostate cancer, and the number of overdiagnoses per prostate cancer death averted. Both costs and benefits were discounted at 3.5% per annum, as per NICE guidelines [28].

**Table 1. Model parameters.**

| Parameter | Value (95% CI)[a] | Distribution Used in Probabilistic Analyses (α, β)[b] | Description | Source |
|---|---|---|---|---|
| *Life table* | | | | |
| RR of prostate-cancer–specific mortality with screening | 0.79 (0.69 to 0.91) | Log-normal [SE: 0.06] | The relative reduction in mortality seen with screening with PSA in the ERSPC. | [8] |
| RR of incidence of prostate cancer with screening | 1.23 (1.03 to 1.48) | Log-normal [SE: 0.18] | Relative increase in the incidence of prostate cancer in the presence of screening with PSA, derived from a meta-analysis of randomised controlled trials of PSA screening. | [18] |
| Proportion overdiagnosed | −0.62 + age × 0.014 | Beta [SE: 0.001] | Derived from linear regression of estimates for the risk of overdiagnosis in 5-year age groups. | [2] |
| RR of advanced cancer at diagnosis if screened | 0.85 (0.72 to 0.99) | Log-normal [SE: 0.07] | Relative decrease in the proportion of cancers that are considered advanced (stages III or IV) if screen-detected, derived from a meta-analysis of randomised controlled trials of PSA screening. | [18] |
| *Utility values* | | | | |
| General population utility | 0.8639 (0.852 to 0.875) | 0.83 + Gamma (4, 0.06) × 0.167 | A yearly utility decline of 0.0048 (0.004 to 0.006) was estimated from linear regression of the mean health state values in 5-year intervals from 45 to 90 against age. | [19] |
| Relative reduction in utility for those with prostate cancer | 0.93, range: [0.88 to 1.0] | 0.88 + Gamma (5, 0.05) × 0.2 | Average over 10 years. Sampled from a right-skewed distribution in probabilistic analysis. | [20] |
| *Costs (GBP£ in 2016 prices)[c,d]* | | | | |
| PSA testing | 11 (7 to 15) | Gamma (33.9, 0.3) | | [11,12] |
| Polygenic risk stratification | 25 (17 to 33) | Gamma (33.9, 0.7) | Empirical estimate calculated from the laboratory costs of genotyping a similar number of SNPs. | |
| Biopsy | 388 (260 to 516) | Gamma (33.9, 11.5) | | [11,21,22] |
| Declined biopsy, but had a PSA ≥ 3.0 ng/ml | 105 (70 to 140) | Gamma (33.9, 3.1) | Individuals who declined biopsy but had a PSA ≥ 3.0 ng/ml were assumed to have one urological appointment.[e] | [11] |
| Staging of diagnosed cancer | 770 (516 to 1,024) | Gamma (33.9, 22.7) | Cost of staging with MRI and an isotope bone scan. | [11,21,22] |
| Active surveillance | 4,341 (2,908 to 5,774) | Gamma (33.9, 128.1) | Average over 10 years, assuming that 55% will go on to have radical therapy during this time period. | [11,23,24] |
| Radical prostatectomy | 8,173 (5,476 to 10,870) | Gamma (33.9, 241.2) | Incorporating the costs of complications and follow-up over 5 years. | [11,21,22] |
| Radical radiotherapy | 5,385 (3,608 to 7,162) | Gamma (33.9, 158.9) | Incorporating the costs of complications and follow-up over 5 years. | [11,21,22] |
| Brachytherapy | 1,527 (1,023 to 2,031) | Gamma (33.9, 45.1) | | [11,21,22] |
| Chemotherapy | 7,426 (4,975 to 9,877) | Gamma (33.9, 219.2) | | [11,21,25] |
| Androgen deprivation therapy | 559 (375 to 744) | Gamma (33.9,16.5) | Derived from the NICE costing statement of its prostate cancer, inflated to 2015–2016 prices, with the addition of 1 urological appointment as follow-up. | [16] |
| Palliation and death from prostate cancer | 6,837 (535 to 20,257[f]) | Gamma (1.8, 3854.9) | Inflated from 2013–2014 estimated costs to the healthcare system in the last 12 months of life. | [26] |

[a]95% CI unless otherwise stated.

[b]α and β parameters shown unless otherwise stated.

[c]In sensitivity analyses, it was assumed that 95% of the costs are likely to vary no more than approximately one-third from the calculated baseline value [27].

[d]All costs are in 2016 GBP£.

[e]Except in sensitivity analysis, all men eligible for biopsy were assumed to have one.

[f]95% credible interval—see Table A in S1 Appendix for further details. **Abbreviations:** CI, confidence interval; ERSPC, European Randomized Study of Screening for Prostate Cancer; NICE, National Institute for Health and Care Excellence; PSA, prostate-specific antigen; RR, relative risk; SE, standard error; SNP, single nucleotide polymorphism.

We calculated incremental cost-effectiveness ratios (ICERs) by dividing the difference in mean costs between the compared scenarios by the difference in mean QALYs, derived from 10,000 probabilistic simulations [29]. The net monetary benefit (NMB) was calculated by subtracting the costs from the product of the QALYs and the willingness-to-pay threshold. We ranked the cost-effectiveness of each screening strategy using different risk thresholds by NMB, using willingness-to-pay thresholds of £20,000 ($26,000) and £30,000 ($39,000) per QALY gained because these reflect the range of thresholds used by NICE in the consideration of the cost-effectiveness of an intervention [28].

### Probabilistic and deterministic sensitivity analyses

To account for uncertainty in the parameters, we ran 10,000 simulations for each screening scenario; the distributions used for each input varied are shown in Table 1. The results presented are probabilistic unless otherwise stated. In sensitivity analyses, we evaluated precision screening starting at the age of 45, 50, and 60; the impact of different levels of screening uptake and adherence; overdiagnosis when adjusted for polygenic risk; and the cost of polygenic testing. We used Python v3.6.5 for all analyses; the code is available at https://github.com/callta/precision_screening_pca. The CHEERS checklist [30] was used in the preparation of this manuscript.

## Results

### Risk of developing prostate cancer

Our analyses show that the background 10-year absolute risk of developing prostate cancer in the absence of screening rose from 2.6% to 7.1% between the ages of 55 and 69 (Fig B in S1 Appendix). Just under half of men aged 55 (49.1%) had a 10-year absolute risk of developing prostate cancer of ≥2% based on age and polygenic risk profile, increasing to 90.6% of men aged 69. The proportion of men by age at or above each 10-year absolute risk threshold is shown in Fig E in S1 Appendix.

### Benefits and harms of screening

In a hypothetical cohort of 4.48 million men, age-based screening led to 39,272 fewer deaths from prostate cancer and to 94,831 overdiagnoses, representing 2.4 overdiagnoses per prostate cancer death averted, as well as 764,446 additional biopsies over 35 years follow-up, by comparison with no screening. The tradeoff between the benefits and harms of precision screening varied by risk threshold (Fig 1, Table 2, Table C in S1 Appendix). In the precision screening cohort, at a 2% 10-year absolute risk threshold, 36,534 deaths from prostate cancer were prevented at the expense of 84,681 overdiagnoses by comparison with no screening. As the risk threshold was raised to 10%, 14,507 deaths were prevented and 26,791 overdiagnosed by comparison with no screening. This represented a drop in the ratio of overdiagnosed cases to prostate cancer deaths prevented from 2.3 to 1.8 at risk thresholds of 2% and 10%, respectively (Fig F in S1 Appendix), and there was a reduction in the number of additional biopsies performed from 652,177 to 150,635 (Table C in S1 Appendix).

### Cost-effectiveness

Age-based screening led to an additional 16,416 QALYs by comparison with no screening, at a cost of £574 million ($746 million) over 35 years, such that the ICER was £34,952 ($45,437) per QALY gained. In 40.7% of simulations, the ICER of age-based compared with no screening was ≤£20,000 ($26,000) per QALY gained.

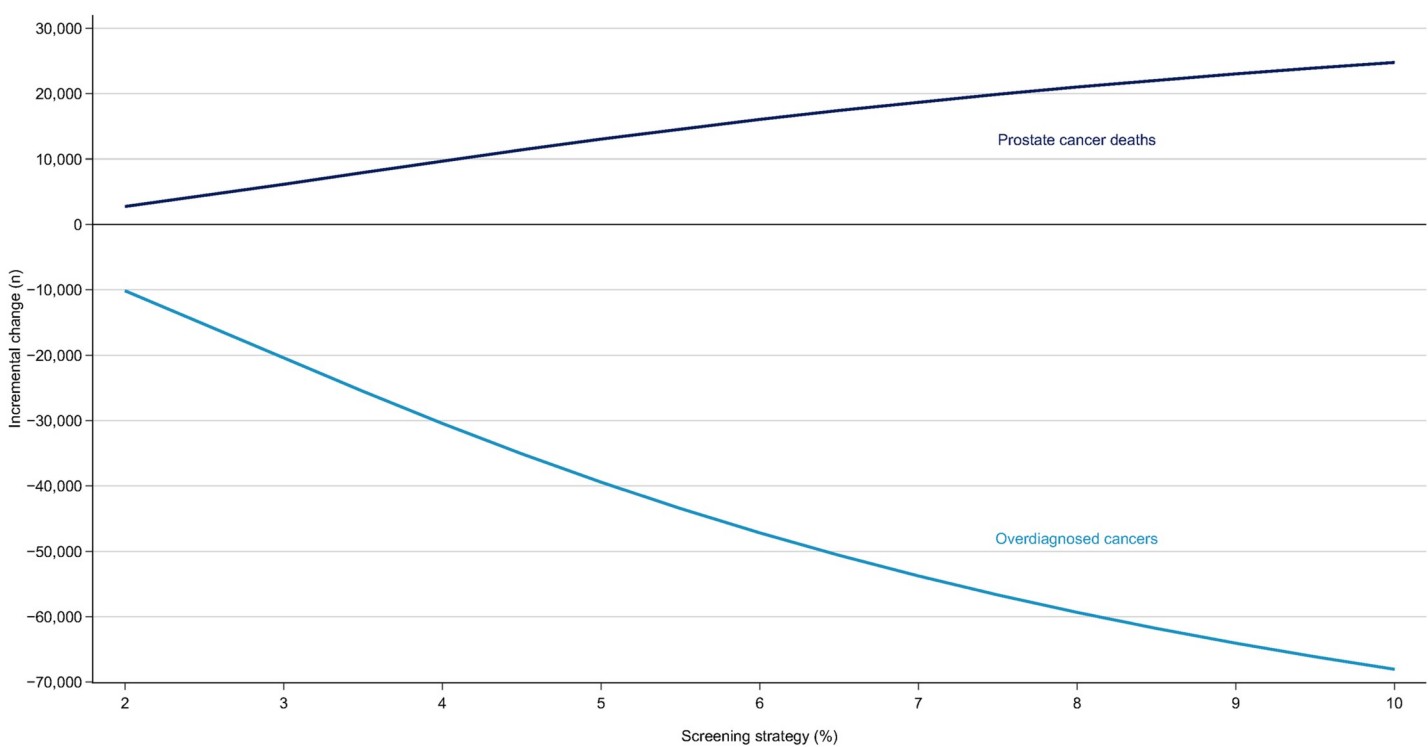

**Fig 1. Overdiagnosed cases and prostate cancer deaths prevented with precision screening as compared to age-based screening.** Results based on 10,000 simulations.

In the precision screening model, below a 4.5% 10-year absolute risk threshold, there was a plateau in the QALYs gained by comparison with no screening that subsequently began to fall as the risk threshold was raised, whilst the costs of precision screening continued to drop as the risk threshold increased (Fig 2). At all 10-year absolute risk thresholds below 10%, precision screening led to a greater number of incremental QALYs gained than age-based screening whilst incurring fewer additional costs at all risk thresholds above 2% (Fig 2).

At a 2% 10-year absolute risk threshold, the ICER was £30,297 ($39,386) per QALY gained. This declined as the risk threshold increased, reaching a plateau at a threshold of approximately 7%, at which point the ICER was £16,755 ($21,781) per QALY gained. Precision screening was cost-effective at a willingness-to-pay threshold of £20,000 ($26,000) per QALY gained compared to no screening at all 10-year absolute risk thresholds above 4.5%. At a risk threshold of 5%, precision screening had a 51.5% probability of being cost-effective and an ICER of £19,598 ($25,478), rising to a 62.5% probability of being cost-effective and an ICER of £14,862 ($19,320) at a threshold of 10%.

## NMB of screening strategies

Comparing all precision and age-based strategies with no screening, the highest NMB at willingness-to-pay thresholds of £20,000 ($26,000) and £30,000 ($39,000) per QALY was seen with precision screening at a 10% and 8% 10-year absolute risk threshold, respectively. Screening strategies by NMB are presented in Fig H in S1 Appendix, whilst the cost-effectiveness acceptability planes, curves, and frontier are shown in Figs I–J in S1 Appendix. Age-based screening had a lower NMB than all precision screening strategies.

**Table 2. Outcomes of age-based and precision screening compared with no screening.**

| Screening Strategy | Prostate Cancer Cases (n) | Difference with No Screening | Overdiagnosed Cases (n) | Deaths from Prostate Cancer (n) | Difference with No Screening | QALYs (n) | Difference with No Screening | Costs (£ Millions) | Difference with No Screening (£ Millions) | ICER (£/QALY Gained) | Cumulative Percentage Screened[a] (%) |
|---|---|---|---|---|---|---|---|---|---|---|---|
| No screening | 537,870 | | N/A | 192,623 | | 46,682,945 | | 2,975 | | | 0 |
| Age-based screening | 644,047 | 106,177 | 94,831 | 153,351 | −39,272 | 46,699,360 | 16,416 | 3,549 | 574 | 34,952 | 100 |
| Precision screening (10-year AR) | | | | | | | | | | | |
| 2.0% | 622,733 | 84,863 | 84,681 | 156,089 | −36,534 | 46,702,653 | 19,709 | 3,572 | 597 | 30,297 | 75.4 |
| 2.5% | 614,230 | 76,360 | 79,620 | 157,723 | −34,900 | 46,703,346 | 20,401 | 3,537 | 562 | 27,542 | 66.7 |
| 3.0% | 606,014 | 68,144 | 74,419 | 159,482 | −33,141 | 46,703,788 | 20,844 | 3,503 | 527 | 25,290 | 58.9 |
| 3.5% | 598,318 | 60,448 | 69,298 | 161,275 | −31,348 | 46,704,012 | 21,067 | 3,469 | 494 | 23,446 | 51.9 |
| 4.0% | 591,244 | 53,375 | 64,384 | 163,045 | −29,578 | 46,704,054 | 21,109 | 3,438 | 463 | 21,924 | 45.8 |
| 4.5% | 584,818 | 46,949 | 59,743 | 164,759 | −27,864 | 46,703,950 | 21,006 | 3,409 | 434 | 20,659 | 40.5 |
| 5.0% | 579,026 | 41,156 | 55,406 | 166,397 | −26,226 | 46,703,733 | 20,788 | 3,383 | 407 | 19,598 | 35.9 |
| 5.5% | 573,830 | 35,960 | 51,379 | 167,947 | −24,676 | 46,703,427 | 20,482 | 3,358 | 383 | 18,704 | 31.9 |
| 6.0% | 569,186 | 31,316 | 47,656 | 169,407 | −23,216 | 46,703,054 | 20,109 | 3,336 | 361 | 17,947 | 28.4 |
| 6.5% | 565,045 | 27,175 | 44,224 | 170,775 | −21,848 | 46,702,631 | 19,686 | 3,316 | 341 | 17,303 | 25.4 |
| 7.0% | 561,358 | 23,488 | 41,065 | 172,055 | −20,568 | 46,702,172 | 19,227 | 3,298 | 322 | 16,755 | 22.7 |
| 7.5% | 558,079 | 20,209 | 38,160 | 173,250 | −19,373 | 46,701,687 | 18,743 | 3,281 | 305 | 16,289 | 20.4 |
| 8.0% | 555,165 | 17,295 | 35,490 | 174,364 | −18,258 | 46,701,187 | 18,242 | 3,265 | 290 | 15,894 | 18.4 |
| 8.5% | 552,577 | 14,707 | 33,036 | 175,403 | −17,220 | 46,700,677 | 17,732 | 3,251 | 276 | 15,560 | 16.6 |
| 9.0% | 550,279 | 12,410 | 30,779 | 176,371 | −16,251 | 46,700,163 | 17,218 | 3,238 | 263 | 15,281 | 15.0 |
| 9.5% | 548,241 | 10,371 | 28,703 | 177,274 | −15,349 | 46,699,649 | 16,704 | 3,227 | 251 | 15,050 | 13.6 |
| 10.0% | 546,432 | 8,562 | 26,791 | 178,116 | −14,507 | 46,699,140 | 16,195 | 3,216 | 241 | 14,862 | 12.3 |

Outcomes shown in hypothetical cohorts of 4.8 million men aged 55 to 69 in England, followed up to age 90. **Abbreviations:** AR, absolute risk; ICER, incremental cost-effectiveness ratio; N/A, not applicable; QALY, quality-adjusted life-year.

[a]Total cumulative proportion eligible for screening by the aged 69. See Fig E in S1 Appendix for an indication of the proportions eligible for precision screening as they age by different risk thresholds. Each ICER represents that specific screening strategy compared with no screening.

## Sensitivity analyses

The results of sensitivity analyses are available in S1 Appendix.

## Discussion

This modelling analysis has shown that precision screening based on age and polygenic risk could reduce overdiagnosis whilst preserving most of the mortality benefits of age-based screening for prostate cancer. At all risk thresholds studied, precision screening had a higher NMB, lower ICER, and fewer overdiagnosed prostate cancers than age-based screening. The cost-effectiveness of screening increases as the risk threshold rises, plateauing at a risk threshold of approximately 7%. However, the greatest QALY gains are at a 10-year absolute risk threshold of 4%. A precision screening programme using a 4% risk threshold would reduce overdiagnosis by one-third, yield more QALYs, and cost less whilst maintaining the benefits of screening. The ideal strategy will depend on both a society's willingness to pay for each QALY gained as well as the tradeoff between benefits and harms considered acceptable both at an individual and population level [31].

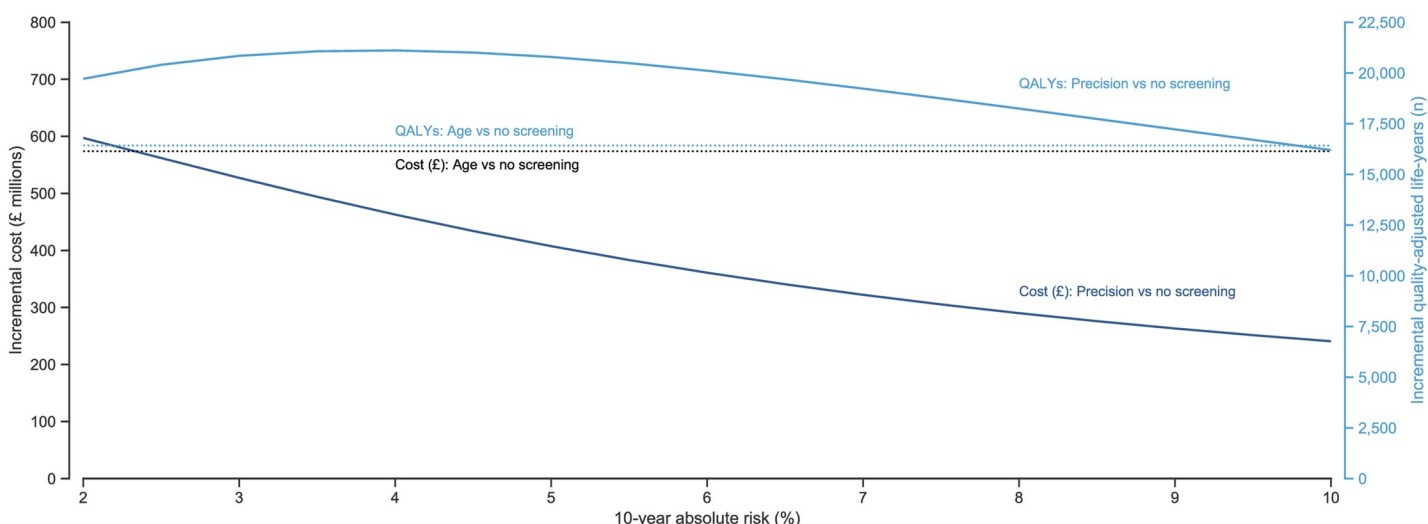

**Fig 2. Incremental cost and QALYs of precision and age-based screening compared with no screening.** Results based on 10,000 simulations. The solid lines describe the incremental costs incurred and QALYs gained of precision screening versus no screening, whilst the dashed lines represent the incremental costs and QALYs of age-based versus no screening. QALY, quality-adjusted life-year.

A plateau is also seen in the cost-effectiveness as the risk threshold for precision screening rises (Table 2, Fig 2). This reflects the fact that fewer deaths are being prevented relative to the increased number of prostate cancer cases, and therefore the greater number of years lived with prostate cancer, as men at higher risk are screened. The incremental QALYs gained with precision screening begin to drop at a 10-year risk threshold above 4%. However, the ICER of precision screening does not begin to plateau until the 10-year absolute risk threshold is raised to 7%. Together, this suggests that a strategy of precision screening at a 10-year absolute risk threshold of between 4% and 7% may provide the most appropriate balance of harms and benefits, considering prostate cancer deaths prevented, cases overdiagnosed, and QALYs gained for the additional costs of screening.

Screening men at a higher risk of prostate cancer lowers the proportion of overdiagnosed cases, the number of additional biopsies performed, and the ratio of overdiagnosed cases to prostate cancer deaths averted. As the risk threshold rose, a smaller proportion of men became eligible for screening. Overdiagnosis dropped as the risk threshold increased. With fewer men screened, there were fewer prostate cancer deaths averted compared to age-based screening. However, the extent of the drop in overdiagnosis was greater than the extent of prostate cancer deaths not prevented, leading to an improvement in the benefit–harm profile as the risk threshold rose (Fig G in S1 Appendix).

In the UK, NICE considers interventions with an ICER of ≤£20,000 per QALY gained as cost-effective, a threshold that was reached with all precision screening strategies above a 5% 10-year absolute risk threshold [28]. A precision screening strategy using a 5% 10-year absolute risk reflects the average risk of developing prostate cancer in men aged 61 in England. A programme employing this strategy would screen 1 in 10 men (11.4%) at the age of 55, rising to just over half (50.5%) by the age of 69. This strategy would reduce overdiagnosis by a 41.6%, yield more QALYs, and cost less, at the expense of 8.5% fewer prostate cancer deaths averted by comparison with age-based screening.

Precision screening for prostate cancer would involve an evolution in screening services, with logistical and ethical implications. Risk tailoring implies that different individuals are invited to screening at different ages, with potential knock-on effects on screening delivery

[32]. In addition, although the disclosure of genetic material to insurance companies is covered under a moratorium in England [33], the broader impact of risk-tailored screening using genetic material on individuals and society requires further research. Finally, the introduction of screening programmes could risk widening health inequalities between both socioeconomic classes and ethnic groups, which can occur as a result of varied uptake amongst different socio-economic strata and ethnicities of screening [34,35]. Research to mitigate this occurrence should be considered alongside prospective studies of polygenic risk-based screening.

In a precision screening programme, in addition to altering the screening start age, the frequency of screening could be varied according to risk. For example, men at higher risk may receive more frequent screening and men at lower risk receive less frequent or no screening. Intensified screening could improve the benefit–harm tradeoffs if the sojourn time—the time it takes to progress from preclinical screen-detectable cancer to clinically detectable cancer—varies with risk level [36]. In the absence of these data in the context of polygenic risk, the impact of varying screening intervals was not estimated.

There have been no comparable studies estimating the impact of polygenic risk-tailored screening in prostate cancer. However, the conclusions reached from our precision and age-based screening models compare favourably with a microsimulation model from the US [37], attesting to the underlying robustness of our model in spite of the differences in model design and assumptions. Using a microsimulation model and a selective treatment strategy involving initial conservative management for those with localised cancer (Gleason score <7 and stage T2a), Roth and colleagues state that quadrennial age-based screening between the ages of 55 and 69 with a PSA cutoff of 3 ng/ml would lead to 30 additional QALYs per 10,000 men screened (37 in this analysis) [37].

## Limitations

Follow-up of individuals ended at the age of 90, reflecting the increasing uncertainty in estimates regarding the incidence and mortality from prostate cancer beyond this age. A life-table approach using aggregate data has been used because of uncertainty in how the natural history of prostate cancer varies by risk. Recent data suggest that there is limited variation in health-related quality of life between stages of prostate cancer, with diminishing utility at higher stages of disease [38]. Because screening leads to a greater proportion of cases detected at early stages, the QALYs recorded in screened cohorts in our model may be underestimated. However, our approach produces similar results to microsimulation models taking into account natural history for age-based screening [37]. Simplifying the model structure in this way minimises the number of underlying assumptions, whilst parameter uncertainty is accounted for with probabilistic sensitivity analyses. The estimates of resource use are limited by the absence of data regarding whether the stage at diagnosis of screen-detected cancers and whether response to treatment varies by polygenic risk.

Because there are no data on how overdiagnosis varies for each percentile of the risk distribution, we have assumed the proportion overdiagnosed is equivalent to that seen with PSA screening alone. In sensitivity analyses in which we assume overdiagnosis to vary with polygenic risk, the balance of benefits and harms of precision screening is substantially improved, suggesting that our model underestimates the potential benefits of precision screening (Fig M in S1 Appendix). We have also assumed that precision screening does not lead to a greater relative risk reduction in mortality than age-based screening. Polygenic hazard scores have been shown to be predictive of aggressive cancer, which potentially could disproportionately benefit from screening [39]. This may lead to a more conservative estimate of the benefit/harm ratio attributed to precision screening in this model.

## Conclusion

Our analyses show that precision screening based on age and polygenic risk profile could improve the benefit/harm tradeoff and cost-effectiveness of a screening programme for prostate cancer. Offering screening to men at a 10-year absolute risk threshold between 4% and 7% could lead to greater QALYs, lower costs, and a 32.1% to 56.7% reduction in overdiagnosis when compared to age-based screening. These findings require verification by a prospective randomised evaluation.

## Supporting information

**S1 CHEERS Checklist. Completed CHEERS checklist of items to report in a health economic evaluation.** CHEERS, Consolidate Health Economic Evaluation Reporting Standards. (DOCX)

**S1 Appendix. Detailed methods and supplementary results.**
(DOCX)

## Author Contributions

**Conceptualization:** Tom Callender, Nora Pashayan.

**Data curation:** Tom Callender.

**Formal analysis:** Tom Callender.

**Investigation:** Tom Callender, Nora Pashayan.

**Methodology:** Tom Callender, Steve Morris, Paul D. P. Pharoah, Nora Pashayan.

**Project administration:** Tom Callender.

**Resources:** Mark Emberton.

**Software:** Tom Callender.

**Supervision:** Nora Pashayan.

**Validation:** Tom Callender, Mark Emberton, Steve Morris, Ros Eeles, Zsofia Kote-Jarai.

**Visualization:** Tom Callender.

**Writing – original draft:** Tom Callender.

**Writing – review & editing:** Tom Callender, Mark Emberton, Steve Morris, Ros Eeles, Zsofia Kote-Jarai, Paul D. P. Pharoah, Nora Pashayan.

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
