## [Decision Letter · Decision Letter 0]

17 Sep 2019

Dear Dr. Callender,

Thank you very much for submitting your manuscript "Polygenic Risk-Tailored Screening for Prostate Cancer: A Benefit-Harm and Cost-Effectiveness Analysis" (PMEDICINE-D-19-02276) for consideration at PLOS Medicine. 

[LINK]

In light of these reviews, I am afraid that we will not be able to accept the manuscript for publication in the journal in its current form, but we would like to consider a revised version that addresses the reviewers' and editors' comments. Obviously we cannot make any decision about publication until we have seen the revised manuscript and your response, and we plan to seek re-review by one or more of the reviewers. 

We expect to receive your revised manuscript by Oct 02 2019 11:59PM. Please email us (plosmedicine@plos.org) if you have any questions or concerns.

We look forward to receiving your revised manuscript. 

Sincerely,

Adya Misra, PhD

Senior Editor 

PLOS Medicine

plosmedicine.org

Abstract-please explain the life table model briefly 

Abstract-please clarify what the analysis from the health service perspective would entail. If this is intention to pay or cost burden, please include this information. 

Abstract-in the methods/findings section please include a sentence about the limitations of your methodology

Abstract-please quantify all results using p-values and 95% confidence intervals

Abstract-in the conclusions section please clarify that these are based on a modelling analysis

Abstract-please tone down the language in the abstract to reflect that there is no clinical application of polygenic risk scores in prostate cancer screening

Author summary- At this stage, we ask that you include a short, non-technical Author Summary of your research to make findings accessible to a wide audience that includes both scientists and non-scientists. The Author Summary should immediately follow the Abstract in your revised manuscript. This text is subject to editorial change and should be distinct from the scientific abstract. Please see our author guidelines for more information: https://journals.plos.org/plosmedicine/s/revising-your-manuscript#loc-author-summary

References-please use square brackets and remove italics from citations. Please use the "Vancouver" style for reference formatting, and see our website for other reference guidelines https://journals.plos.org/plosmedicine/s/submission-guidelines#loc-references

Introduction-Please include a sentence about PSA testing, why there is a substantial risk of false positive diagnosis. The second sentence requires clarification, please revise or consider removing.

Introduction-please introduce polygenic risk at first view and explain how this may enable tailored screening

Methods-Please ensure that the correct link to the Python code has been provided as the current link is not working

Methods-Please provide paragraph or line numbers in the CHEERS checklist as page numbers are likely to change

Methods-Please briefly explain the life table model

Discussion- please include something like “according to our modelling analysis...” at the start of the section

Last sentence of the conclusion- please revise to clarify that and RCT would be essential, instead of valuable 

Overall please use careful language to indicate that the results are based on a model 

Comments from the reviewers:

Reviewer #1: In this article, the authors use a life-table based model to estimate the benefits, harms and cost-effectiveness of a program that screens men in the U.K. for prostate cancer based on their risk (based on age and polygenic profile) versus age-based screening and no screening.

The authors found that risk-based screening using a risk threshold for screening commencement of 5% or over was cost-effective (using a willingness-to-pay threshold of £20,000 per QALY) when compared with age-based screening.

Overall this is an interesting study that is addressing the important question of how to optimize prostate cancer screening and limit the harms associated with the high proportion of overdiagnosed cases. However I believe that the modelling approach used in this paper is not the most appropriate and restricts the possible characteristics of a risk-based screening program that can be explored and evaluated.

For example, risk-based screening programs using different screening intervals and/or PSA thresholds cannot be evaluated because the relative risk of prostate cancer mortality in the presence of screening is taken directly from the ERSPC trial and therefore the model in this manuscript is constrained to do what was done in the trial.

Well calibrated and validated microsimulation models that use a more detailed natural history of prostate cancer are a better alternative to perform a comprehensive evaluation of a risk-based screening program for prostate cancer. Such models have been developed for some countries (e.g. the United States and the Netherlands - see https://cisnet.cancer.gov/prostate/) and have been successfully used in several evaluations, some of which are cited in the manuscript. This alternative approach would for example

* Allow different screening intervals, PSA thresholds, starting/stopping ages for screening to be evaluated

* Evaluate emerging triage technologies (i.e. not all men with PSA > 3ng/ml would be sent directly to biopsy - this could potentially further reduce overdiagnosis);

* Directly assert from the simulation whether a prostate cancer case is an overdiagnosed case as opposed to using a separate model for this classification;

* Better model the introduction of screening and the transitory effects that could occur.

A microsimulation model of prostate cancer that accounted for individual risk profile would be an interesting and worthwhile addition to the current status of prostate cancer modelling research.

.

The authors rightly state in the appendix that a life table based model is simpler and limits the number of unknown parameters when compared with microsimulation models. However there are various methods that are regularly used to parametrize microsimulation models together with uncertainty analysis methods that look at how sensitive the outcomes are to the fitted parameter sets.

Lastly, I tried to access the Python code in https://github.com/callta/precision_screening_pca but the repository was empty (tried to access this on 2019-08-30).

Reviewer #2: Callender et al provide a timely and interesting comparative effectiveness analysis of prostate cancer screening. They show convincingly that a precision approach offers substantial benefits in terms of reduced costs per QALY and reduced over-diagnosis, at given thresholds of lives lost to prostate cancer. They argue that risk-based screening based instead of on age, on combined genetics and age-adjusted 10-year risk of 4% to 7% provides the optimal balance that is likely to meet willingness-to-pay recommendations for the UK.

There is however one important element of the study that is missing, and that is any evaluation of the performance of polygenic risk assessment. The main reference [7] given for justifying the parameter they include in their modeling is a 2002 paper on breast cancer, and the only other reference to a PRS is [28] in the last sentence of the manuscript. In addition to that study, a quick search identified two more (PMID: 30366021, PMID: 29892016) with polygenic scores incorporating from 7 to 147 SNPs, and it should now be possible based on the data in the latter study to generate a truly genome-wide score based on millions of variants. I think it is essential to consider in this paper the impact of the likely improvement in diagnostic utility of PRS as they explain more of the (genetic) variance of disease. Relative risks in the top percentiles are now approaching 3X greater than the median; and similarly at the low end of the scale it should be possible to identify individuals whose negative predictive value is such that it provides even more utility than positive predictive value. Presumably a composite score combining age, polygenic risk, and other prostate cancer risk factors (eg family history, prior PSA tests), and ancestry will do even better. 

Several recent reviews have addressed the use of PRS in clinical contexts. This paper is a really nice example of combining such assessments with real-world prediction of utility. It would be even stronger with commentary on the impact of the discriminatory power of the PRS.

Reviewer #3: This is a promising paper but it needs a few major modifications before finer points can even be assessed.

The first major issue is that it is unclear that the model has necessarily converged. Only 10,000 model runs were used and not all of the line-by-line differences in Table 2 were in the expected directions. As a result, I suspect that this hasn't converged, and that results found may be slightly different were a large number of have been used and convergence achieved. (If I have misunderstood and this is the authors' expected directions, greater clarity over expectations would be helpful.)

The abstract of the paper is very poor in comparison to the elements of the analysis. Given that the paper's results would find that age-based screening is not cost-effective (ICER at 34k per QALY) vs no-screening, the comparison of age-based and precision screening is largely irrelevant. The comparisons against "no screening" in Table 2 are better but this should be incremental. 

This lack of a truly incremental analysis is the biggest (and most telling) flaw in the paper. The paper does not identify a clear enough decision problem, start with a true decision problem, or present results that answer a clear decision problem. Table 2 provides really helpful information for decision making but does not do so in a very systematic (or indeed a truly incremental) fashion. If each of these lines were taken to be a separate decision option (as they are) then this would be much stronger --- here, with back of the envelope calculations (and removing the options that appear to be dominated or extended dominated), it looks to me that the most cost-effective option is to provide screening a risk threshold of 10% with a CE threshold of 20k per QALY and at 8% for a CE threshold of 30k per QALY. I can identify these figures by using Table 2 and calculating ICERs ... this is what the paper should present (it does indirectly where NMB is presented but only as an aside in the discussion, rather than as the results).

 QALYs Incr QALYs Costs Incr Costs ICER

NS 46,682,945 2,975,391,145 

10% 46,699,140 16,195 3,216,074,093 240,682,948 14,862

9.50% 46,699,649 509 3,226,784,314 10,710,221 21,042

9.00% 46,700,163 514 3,238,490,220 11,705,906 22,774

8.50% 46,700,677 514 3,251,296,757 12,806,537 24,915

8.00% 46,701,187 510 3,265,319,785 14,023,028 27,496

7.50% 46,701,687 500 3,280,686,485 15,366,700 30,733

7.00% 46,702,172 485 3,297,535,256 16,848,771 34,740

6.50% 46,702,631 459 3,316,014,724 18,479,468 40,260

6.00% 46,703,054 423 3,336,281,231 20,266,507 47,911

5.50% 46,703,427 373 3,358,493,767 22,212,536 59,551

5.00% 46,703,733 306 3,382,804,647 24,310,880 79,447

4.50% 46,703,950 217 3,409,343,088 26,538,441 122,297

4.00% 46,704,054 104 3,438,186,910 28,843,822 277,344

Note that on the results provided, age-based screening is dominated and doesn't appear in a sensible table of overall results (but would appear in a CEAC).

In an ideal world, the paper should be reframed around this type of view, since it's the standard approach for economic evaluations to consider all relevant options together and to do so incrementally. The paper should also consider some additional points between 9.5% and 10%, and between 7.5% and 8%, in order to optimise this targetted figure a little more closely.

Once this is done then this should really be presented on a CEAC and with a cost-effectiveness frontier; both analyses need to appear as they're standard practice - but neither do at present.

Given all this, the authors suggestion at the beginning of the discussion that a threshold of 4% to 7% is optimal is utterly indefensible on economic grounds when this purports to talk about a willingness to pay of 20k or 30k per QALY. At 4%, their own results suggest that this is cost-effective only if society is willing to pay in excess of 250k per QALY. The authors have, it appears, done a nice piece or work here but don't appear to have correctly used the methods that they state --- this is, however, very easily fixed!

Reviewer #4: I confine my remarks to statistical aspects of this paper.

These were generally fine, but I do have a couple of suggestions.

p 5 - Instead of four yearly put "four times a year" or "every four years".

I'm a little confused as to the data. If it was simulated (top of p. 5) then how could people be invited to do something (middle of p. 5)?

Table 1 - I can't comment on the values chosen, but the distributions seem reasonable.

Table 2 - maybe put QALY in millions and costs in thousands? 

Figure 1 a) I am not sure how the bottom two rows mesh with the relative risk rows. It seems like there would be age based screening and no screening rates for each 10 year risk. b) I would make this into a line graph with risk on the x axis, and frequency on the y axis, with lines as needed. 

Figure 2. Dual axis graphs are not good (see the work of William Cleveland). Either make this into two graphs or show e.g. the ratio of QALY to cost.

Peter Flom

[LINK]

---

## [Decision Letter · Decision Letter 1]

1 Nov 2019

Dear Dr. Callender,

Thank you very much for re-submitting your manuscript "Polygenic Risk-Tailored Screening for Prostate Cancer: A Benefit-Harm and Cost-Effectiveness Analysis" (PMEDICINE-D-19-02276R1) for review by PLOS Medicine.

I have discussed the paper with my colleagues and the academic editor and it was also seen again by xxx reviewers. I am pleased to say that provided the remaining editorial and production issues are dealt with we are planning to accept the paper for publication in the journal.

[LINK]

We look forward to receiving the revised manuscript by Nov 08 2019 11:59PM. 

Sincerely,

Adya Misra, PhD

Senior Editor 

PLOS Medicine

plosmedicine.org

Requests from Editors:

Title- Suggest adding a study descriptor. “A benefit-harm and cost effectiveness analysis of polygenic risk-tailored screening for Prostate Cancer: a modelling study” or similar 

Please add p values to the 95%CIs throughout

Please add a sentence of limitations in the final sentence if the ‘methods and findings’ section of the abstract The R2R mentions this has been added but we are unable to see it. 

I think the author summary overreaches what can be concluded from the results…”Genome wide association studies have identified more than 160 common genetic variants that, when combined together as a polygenic risk score, can be used to develop a tailored screening programme for prostate cancer.”

And "Based on this model, we show that a polygenic risk-tailored screening programme would reduce overdiagnosis, maintain the mortality benefits of age-based screening, and improve the cost-effectiveness of a screening programme for prostate cancer"

Please tone down the conclusions throughout, clarifying the results are based on a model

The square brackets are in the wrong place (after full stop)

The data statement, contributor and so on can be removed from the main text as they get pulled in automatically from EM (page 21 etc)

Abstract background and Introduction-last sentence suggests this is a clinical trial, please revise to clarify a modelling approach was used 

Results first sentence should begin with “according to our model” or “our analyses show” or similar 

Comments from Reviewers:

Reviewer #1: I would like to thank the authors for their replies. However, these did not change my believe that the life table modelling approach used in this paper is not the most appropriate to explore the effectiveness and cost-effectiveness of risk-based screening programs for prostate cancer.

Reviewer #2: I agree that current genome-wide polygenic risk scores are only marginally better than those based on GWAS-significant SNPs. However, my point was that these will continue to improve as sample sizes increase and the variance explained grows. I feel that there is still more that you could comment on the impact of the PRS performance on your modeling now and in the future. But it just a suggestion.

Reviewer #4: The authors have addressed my concerns and I now recommend publication.

Peter Flom

[LINK]

---

## [Editor Report · Decision Letter 2]

19 Nov 2019

Dear Dr. Callender, 

On behalf of my colleagues and the academic editor, Dr. Steven D. Shapiro, I am delighted to inform you that your manuscript entitled "Polygenic risk-tailored screening for prostate cancer: A benefit-harm and cost-effectiveness modelling study" (PMEDICINE-D-19-02276R2) has been accepted for publication in PLOS Medicine. 

PRODUCTION PROCESS

PRESS

PROFILE INFORMATION

Thank you again for submitting the manuscript to PLOS Medicine. We look forward to publishing it. 

Best wishes, 

Adya Misra, PhD

Senior Editor 

PLOS Medicine

plosmedicine.org